# Empirical Assessment of Sequence-Based Predictions of Intrinsically Disordered Regions Involved in Phase Separation

**DOI:** 10.3390/biom15081079

**Published:** 2025-07-25

**Authors:** Xuantai Wu, Kui Wang, Gang Hu, Lukasz Kurgan

**Affiliations:** 1School of Mathematical Sciences and LPMC, Nankai University, Tianjin 300071, China; jasonwxt@mail.nankai.edu.cn; 2NITFID, School of Statistics and Data Science, LPMC and KLMDASR, Nankai University, Tianjin 300071, China; wangkui@nankai.edu.cn (K.W.); huggs@nankai.edu.cn (G.H.); 3Department of Computer Science, Virginia Commonwealth University, Richmond, VA 23284, USA

**Keywords:** phase separation, membrane-less organelles, biomolecular condensates, intrinsic disorder, prediction, assessment

## Abstract

Phase separation processes facilitate the formation of membrane-less organelles and involve interactions within structured domains and intrinsically disordered regions (IDRs) in protein sequences. The literature suggests that the involvement of proteins in phase separation can be predicted from their sequences, leading to the development of over 30 computational predictors. We focused on intrinsic disorder due to its fundamental role in related diseases, and because recent analysis has shown that phase separation can be accurately predicted for structured proteins. We evaluated eight representative amino acid-level predictors of phase separation, capable of identifying phase-separating IDRs, using a well-annotated, low-similarity test dataset under two complementary evaluation scenarios. Several methods generate accurate predictions in the easier scenario that includes both structured and disordered sequences. However, we demonstrate that modern disorder predictors perform equally well in this scenario by effectively differentiating phase-separating IDRs from structured regions. In the second, more challenging scenario—considering only predictions in disordered regions—disorder predictors underperform, and most phase separation predictors produce only modestly accurate results. Moreover, some predictors are broadly biased to classify disordered residues as phase-separating, which results in low predictive performance in this scenario. Finally, we recommend PSPHunter as the most accurate tool for identifying phase-separating IDRs in both scenarios.

## 1. Introduction

The compartmentalization of cells, which encompasses membrane-enclosed organelles and membrane-less organelles [1], facilitates spatiotemporal organization and regulation of numerous cellular processes. The membrane-less organelles are biomolecular condensates that are established via a phase separation process [1,2]. For proteins, this process is driven by interactions of both folded domains and intrinsically disordered sequence regions [1,3,4,5,6]. The dysregulation and malfunction of the phase separation processes are linked to several diseases [7], including cancers and neurodegeneration [8,9,10], motivating the development of novel tools for identifying and studying biomolecular condensates.

Research showed that involvement of proteins in the phase separation can be predicted from their sequences [11,12], which motivated the development of numerous computational predictors [12,13,14,15]. While most of these tools identify proteins involved in phase separation [12,13,14], several recent methods provide more granular predictions of the condensate-driving intrinsically disordered regions (IDRs) in protein sequences [15,16]. The ability to predict the phase separation-associated IDRs is supported by an observation that they are a distinct class of IDRs [17]. Past research suggests a few factors that can be potentially used to differentiate phase separation-associated IDRs from other types of IDRs, such as the polymer scaling exponent [18], propensity for binding [19], enrichment in beta-turns [18], and depletion in alpha-helices [17]. Prediction of these regions requires the generation of the propensities for phase separation activity for individual amino acids in the input sequence, in contrast to the protein-level methods that produce this propensity for the entire protein sequence. The early computational predictors of the phase separation, such as PLAAC [20] and catGranule [21], were developed in the mid-2010s, and these and subsequent efforts were documented in a few comparative surveys [12,13,14,15]. The two early comparative studies were released in 2019 and 2021, and they both assessed seven protein-level predictors [12,13]. These works performed a comparison on test proteins that were not evaluated for their potential similarity to the phase-separating proteins that were used to train/design the considered predictors. This may result in an inflated measurement of predictive performance because highly similar proteins can be easy to predict, and it potentially favors methods that use more similar training datasets. One of the more recent comparative studies evaluated 11 methods on the test dataset that shares <50% sequence similarity with the training proteins, but it also focused solely on the protein-level predictions [14].

To the best of our knowledge, the only comparative analysis of the amino acid-level predictors was published in 2023 by Gsponer and colleagues [15]. They evaluated five methods, including PScore [22], LLPhyScore [23], catGranule [21] ParSe [18], and FuzDrop [19,24] (the newest tool was from 2022) on the test dataset of 46 proteins [15]. They considered two assessment scenarios: prediction of phase separation associated IDRs among structured and disordered amino acids vs. among disordered regions. The authors found that the best of these five methods provides modest predictive quality in the harder second scenario, with the Area Under the ROC curve (AUROC) of 0.667 [15]. However, the authors used predicted annotations of disordered regions as the ground truth, which could impact the reliability of their observations. Moreover, like in the two earlier protein-level assessments, they did not screen the test proteins for similarity to the phase separation-associated proteins/IDRs that were used to train the evaluated predictors.

We investigated whether current amino acid-level methods can accurately predict phase separation-associated IDRs. This follows recent studies that investigated the quality of predicting different classes of IDRs, such as linkers and binding regions [25,26,27,28,29]. Our focus on the intrinsic disorder is motivated by four factors. First is the fundamental role of the intrinsic disorder in the phase separation processes, including in the stress granule assembly, formation of the membrane-less organelles, and implementation of the innate immune response [4,5,6,30]. Second is the association with numerous pathologies, such as neurodegenerative diseases, cancers, and inflammation-related diseases [31,32,33,34,35]. Third are the abovementioned drawbacks of the previous comparative study and the modest predictive quality it reported [15], which call for an updated and improved evaluation. Lastly, recent empirical analysis already revealed that phase separation can be accurately predicted for the structured proteins [36], which points to little value in reassessing predictions for these proteins. We addressed shortcomings of the evaluation in ref. [15] by taking advantage of the recently released curated collection of the phase separation-associated IDRs [37]. More specifically, we relied on the experimentally annotated intrinsic disorder and ensured that the test proteins are dissimilar (<30% sequence similarity) to the phase separation-affiliated proteins that were utilized to train the evaluated predictors. Moreover, we covered recently released tools, such as Seq2Phase [38] and PSPHunter [39], as well as the methods that were originally evaluated in ref. [15]. Altogether, we performed an independent evaluation (i.e., we were not involved in the development of the phase separation predictors) for a representative collection of predictors on the well-designed test dataset to provide insightful analysis of their ability to provide accurate predictions of IDRs that are involved in phase separation.

## 2. Materials and Methods

### 2.1. Survey and Selection of Predictors for Comparative Analysis

We identified a comprehensive collection of the sequence-based predictors of phase separation based on the past surveys of this area [12,13,14,15] and manual analysis of relevant queries in PubMed and Google Scholar. Altogether, we found 31 phase separation predictors that we list in Table 1. The earliest method was published in 2014 [20]. Nearly 75% of these methods (23 out of 31) were published since 2021, and 30% (nine methods) were released in 2024. This shows strong and recent interest in developing these predictive tools.

We analyzed their availability to the end users, as this was empirically shown to substantially affect their impact [62]. These predictive tools are available as code and/or web servers. The former mode of availability targets more computer-savvy users who are able to install and use these methods on their computing infrastructure. On the positive side, this facilitates large-scale predictions and embedding these tools into broader bioinformatics platforms. The web servers cater to a broader population of users by providing predictions via a web interface and without the need to install software locally. On the negative side, they usually limit the input size to one or a few proteins, which means that users must run multiple jobs to predict larger collections of proteins. About half of these tools (15/31 = 48%) are available as web servers vs. 58% as code (18/31), and only a quarter (8/31 = 26%) offer both options. Moreover, the authors of the five predictors did not provide either code or a server. The relatively low rate of server availability is concerning since this mode was shown to boost use and citations [62].

Importantly, we investigated which methods produce amino acid-level vs. protein-level predictions and reported this in Table 1. We found that only nine of the 31 tools generate the amino acid-level scores that are necessary to identify sequence regions involved in the phase separation, while the remaining 22 tools produce protein-level scores that cannot be used to find the phase separation-associated IDRs. Table 1 summarizes the predictive models used by these nine methods and provides URLs to their code and/or web servers. The seven earlier methods, which include PLAAC [20], catGranule [21], PScore [22], FuzDrop [19], LLPhyScore [23], ParSe [18], and ParSe v2 [55], rely on scoring functions (i.e., linear regressions or similar weighted combinations of a few input features) to make predictions. However, they differ in terms of their predictive inputs and how they process them, where the latter typically involves using sequence windows/regions of varying sizes. The inputs include the sequence, composition (fraction) of specific amino acid types (e.g., charged amino acids), hydrophobicity, predicted intrinsic disorder, propensity for helix and strand conformations, and various types of interactions including pi-pi, residue-water, and residue-RNA. The two most recent tools, Seq2Phase [38] and PSPHunter [39], utilize more sophisticated machine learning models, such as random forest, support vector machine and a feedforward neural network, and a broader selection of inputs that cover sequence conservation, sequence embeddings generated by a popular protein language model, presence of low complexity regions, and predictions of intrinsic disorder, secondary structure, solvent accessibility, protein-DNA and protein-RNA binding, and posttranslational modifications. Our analysis aims to determine whether these differences in models and inputs lead to improvements in predictive performance.

Next, we selected tools for inclusion in our comparative evaluation. We were limited to the nine predictors that produce amino acid-level results and checked whether they are available as the web servers and/or source code, which we could use to make predictions on our new test dataset. All nine tools are available, and we removed ParSe since we included its newer and more accurate version, ParSe v2 [55]. Consequently, we evaluated eight methods (chronologically): PLAAC [20], catGranule [21], PScore [22], FuzDrop [19], LLPhyScore [23], ParSe v2 [55], Seq2Phase [38], and PSPHunter [39]. These methods encompass the five tools that were assessed in the previous study in ref. [15], except that we used the newer version of the ParSe predictor (ParSe v2), and we also added two recent tools from 2024. We collected predictions from these eight methods using either their web servers (catGranule, FuzDrop, ParSe v2, and PSPHunter) or the author-provided code that we installed and ran locally (LLPhyScore, PLAAC, PScore, and Seq2Phase).

### 2.2. Collection and Annotation of Test Dataset

Our high-quality test dataset relies on the experimental annotations of phase-separating regions and intrinsic disorder, and we ensured that the test proteins share low levels of similarity with phase-separating training proteins that were used to develop the eight tested predictors. Moreover, we balanced the dataset to represent three distinct types of proteins that include phase-separating IDRs, IDRs with other (non-phase-separating) functions, and structured proteins. This allowed us to evaluate two distinct practical scenarios: (1) predicting phase-separating IDRs among disordered residues; and (2) predicting phase-separating IDRs among both structured and disordered residues.

We collected proteins with the experimentally validated phase-separating IDRs from the two databases that provide the amino acid level annotations, DisProt [37] and PhaSepDB [63]. In particular, DisProt ver. 8 introduced annotations of the phase-separating IDRs [64] and the corresponding curated dataset, which we used to collect our test data, was released as part of DisProt ver. 9.5 [37]. We also collected phase-separating regions from PhaSepDB ver. 2.1. We excluded regions with invalid or unrecognizable format (e.g., ending location greater than the length of the protein and unspecified position in the sequence) and included only those located in the disordered regions, which we annotated by using experimentally validated data from DisProt. We merged the phase-separating IDRs that were annotated for the same protein across the two sources.

We obtained proteins with the non-phase-separating IDRs from DisProt version 9.5. To minimize the risk that these regions could be involved in the phase separation, we only selected proteins with IDRs that have at least two experimentally validated disorder function annotations and where none of these annotations is related to the phase separation. This is a marked improvement over the recent comparative study in ref. [15] where the predicted disorder that was not required to be annotated with a function was utilized.

We acquired fully structured proteins, which by definition do not include phase-separating IDRs, from the Protein Data Bank (PDB) [65]. More specifically, we extracted proteins with high-resolution X-ray structures (≤2 Å) that involve a single chain that is at least 140 amino acids long (the minimum sequence length required to run PScore), no ligands, and that have at least 99% of structurally resolved residues. These requirements aim to avoid using structures of complexes where the corresponding interaction interface could be potentially disordered in isolation and could have a phase separation-associated function. We also ensured that the resulting sequences share low sequence similarity (<30%) with each other and map to full UniProt sequences with over 99% coverage, i.e., the entire protein sequence is structured.

Next, we clustered all our collected proteins together with the phase separation-associated proteins that were used to train the selected eight predictors. We obtained the phase-separating training proteins for the eight predictors from the Appendix A (for catGranule, FuzDrop, LLPhyScore, ParSe, Pscore, Seq2Phase), website (for PSPHunter), and manuscript (for PLAAC). We performed clustering using BLASTCLUST version 2.2.26 [66] with the 0.3 sequence similarity threshold (*blastclust-S 30* command) and defaults for the other parameters, which set the sequence length coverage at 0.9 and result in the use of BLOSUM62 matrix to quantify similarity. We generated 703 clusters from the 1443 proteins that we processed. We collected the test proteins with the phase-separating IDRs only from the clusters that did not contain the training proteins, and the test proteins with the non-phase-separating IDRs and structured proteins only from the clusters that did not include the training proteins and the phase-separating proteins. This ensures that the selected test proteins have below 30% similar sequences compared to the training proteins of the eight predictors and minimizes the likelihood that the selected test proteins that are structured and that include non-phase-separating IDRs have (unannotated) phase-separating IDRs. Consequently, we managed to extract 16 phase-separating test proteins, and we correspondingly sampled 16 proteins from each of the two other groups. We randomly selected 16 structured proteins, and we selected 16 proteins with non-phase-separating IDRs to match the distribution of disorder (number of fully disordered proteins and proteins with different numbers of IDRs) to the 16 phase-separating test proteins. The latter aims to ensure that the potential ability of predictors to identify phase-separating IDRs is not driven by an unusual pattern of disorder amounts in these proteins.

To sum up, we extracted 48 test proteins that uniformly cover proteins with phase-separating IDRs, non-phase-separating IDRs, and structured proteins, and which share <30% sequence similarity to the training phase-separating proteins. The 30% similarity cutoff is stricter (more robust) than the cutoff of 50% used in ref. [14]. This is motivated by the fact that comparative evaluations typically rely on such stricter cutoffs at around 30% [29,67,68,69], given that sequence alignment does not produce accurate results at these levels of similarity, and to ensure that all methods are evaluated fairly. Our test dataset has a similar size to the test data of 46 proteins from the ref. [15], the only published study that performed evaluations at the amino acid level, and fixes key issues of that test data, such as the use of predicted disorder as ground truth, and potentially high similarity to the training data [15]. While our assessment is at the amino acid level and the test set covers about 23 thousand amino acids, we acknowledge that the number of proteins included is rather limited, which can adversely affect the reliability of our observations. This limitation stems from the relatively small number of experimentally annotated phase-separating IDRs that are currently available in DisProt [37] and PhaSepDB [63], especially after we factored in the removal of test data based on similarity to the training datasets. The test dataset, including the annotations of the phase-separating IDRs, other types of IDRs, and structured residues, is available in the Supplement.

### 2.3. Assessment of Predictive Performance

The selected predictors generate a numeric propensity for phase separation for individual amino acids in the input protein sequence. We evaluated the predictive quality of these propensities using two popular metrics, Area Under the ROC Curve (AUROC), and Area Under the Precision-Recall curve (AUPRC). The propensities are typically converted into binary state predictions, phase-separating vs. non-phase-separating, where residues with propensities higher than a threshold are considered as phase-separating, and otherwise they are assumed as non-phase-separating. We set the threshold for each predictor to the value that generates the correct number of phase-separating residues across the entire test dataset (i.e., we used a single threshold for each predictor). We did that for comparative evaluation purposes to facilitate a consistent comparison between results produced by different methods, and to ensure that the binary state predictions are balanced. We note that individual predictors typically generate binary state predictions using their own thresholds that are optimized by the authors, and which should be utilized by the end users. However, such predictions could not be reliably compared across methods since they would result in vastly different rates of predictions of phase-separating residues, which in turn would affect the ability to directly compare values of the considered metrics of predictive performance. We assessed the binary predictions using two metrics:$$F1\;=\;\frac{2\times Recall\times Precision}{Recall+Precision}$$$$\mathrm{MCC}=\frac{TP\times TN-FP\times FN}{\sqrt{\left(TP+FN\right)\left(TP+FP\right)\left(TN+FP\right)\left(TN+FN\right)}}$$
where TP and TN are the numbers of the correctly predicted phase-separating and non-phase-separating residues, respectively, FP is the number of the non-phase-separating residues incorrectly predicted as phase-separating, FN is the number of the phase-separating residues incorrectly predicted as non-phase-separating, where precision = TP/(TP + FP) and recall = TP/(TP + FN). Three of the four prior assessments of predictors of phase separation relied solely on the AUROC values [12,13,15], while the other assessment also utilized MCC and F1 scores [14]. The use of additional metrics allowed us to provide more robust conclusions and was motivated by their inclusion in related evaluations of protein structure or function predictions [25,26,27,28].

We performed tests of statistical significance of differences in predictive performance that compared the considered methods against two points of reference, the most accurate method and a random baseline that generates random real-valued numbers between 0 and 1. These tests evaluate whether these differences are consistent across different datasets. To this end, we randomly sampled half of the test proteins, repeated it 10 times, and compared the resulting 10 paired results. If these measurements are normal (based on the Anderson-Darling normality test at the 0.05 significance) then we used the paired *t*-test to assess the statistical significance; otherwise, we utilized the paired Wilcoxon rank test. We assume that the differences are statistically significant if the corresponding *p*-values < 0.05.

## 3. Results

Similarly to ref. [15], we considered two evaluation scenarios: (1) prediction of phase separation associated IDRs in the protein sequences (i.e., among structured and disordered amino acids); and (2) prediction of phase separation associated IDRs among disordered amino acids. The first scenario is relatively easier since the disordered amino acids, which include the phase-separation associated IDRs, are very distinct from the structured residues [70,71]. The second scenario is more challenging and arguably reveals the true value of the underlying predictors, especially since the disordered residues can be accurately predicted by many easily accessible disorder predictors [26,27,28,69,72,73]. Correspondingly, we annotated each amino acid in the test proteins into one of the four types: P-type (located in the experimentally annotated IDRs associated with Phase separation); F-type (in the experimentally annotated IDRs with at least two non-phase separating Functions); S-type (in the experimentally annotated Structured proteins); and residues that could not be unambiguously categorized into one of the other three groups (e.g., they lack disorder, structure and/or function annotations). We excluded the latter group of amino acids from the assessment to ensure that only well-annotated data are used. Correspondingly, we used all well-annotated disordered and structured residues, including P, F, and S in the first scenario, vs. the P and F residues in the second scenario. Moreover, following recent related studies [25,26], we performed random sampling that generates equal proportions of the phase separation-related residues to facilitate direct comparison of results between the two scenarios. More specifically, we randomly sampled equal numbers of P and F residues in the second scenario, and the combined number of F and S residues was equal to the number of P residues for the first scenario, i.e., F + S = P. We used predictions from different methods for the same collection of sampled residues that were pooled across each of the 10 test protein sets, consisting of half of the test dataset, which we used to perform analysis of the statistical significance of differences.

### 3.1. Prediction of Phase Separation in Structured and Disordered Residues

Table 2 summarizes results for the first scenario, where we were predicting phase separation-associated IDRs among well-annotated structured and disordered residues on the low-similarity test dataset. We assessed the eight representative phase separation predictors and compared them against a random baseline and two recently released disorder predictors, AIUPred [74] and flDPnn [75,76]. These two disorder predictors secured accurate results in the community-run Critical Assessment of Intrinsic Disorder (CAID) experiments [27,28] and are popular and conveniently available as web servers and code. We utilize outputs generated by these two methods (i.e., their predictions of intrinsic disorder) as additional baselines that can be used to investigate whether the phase separation predictors would outperform the disorder predictors in the context of our study, which focuses on the intrinsic disorder. We note that the disorder predictors are expected to perform better than random based on scenario 1, where they should be capable of distinguishing phase separation-associated IDRs from structured residues. However, disorder predictors should perform only on par with a random baseline in scenario 2, given that they should not be able to distinguish phase separation-associated IDRs from other types of IDRs.

We found that all methods are statistically more accurate than a random baseline across the four metrics of predictive performance (*p*-values < 0.05; Table 2). Moreover, several methods, which include PSPHunter, FuzDrop, LLPhyScore, Seq2Phase, and AIUPred, provide accurate predictions, with AUROCs ≥ 0.81 and AUPRCs ≥ 0.75. Interestingly, as expected, both disorder predictors generate accurate results, although they were not designed to identify the phase-separating IDRs. This is because they accurately differentiate between structured and disordered residues, which indirectly separates the phase-separating IDRs from the structured residues. However, these two tools should perform worse when making predictions for full sequences of disordered proteins (i.e., unsampled data) where we can expect that the number of P-type residues should be lower than F-type residues (i.e., the phase separation-associated IDRs are a relatively small subset of all IDRs). Several predictors of phase separation, such as FuzDrop, LLPhyScore, Seq2Phase, and PLAAC, match the predictive quality of the disorder predictors, which suggests that they can accurately distinguish disorder from structure and/or identify phase-separating regions among the disordered and structured residues. We used the second scenario to further investigate this aspect. The most accurate PSPHunter secures the highest AUROC, AUPRC, MCC, and F1 values, which are statistically better than the results of all other tools, including the two disorder predictors (*p*-values < 0.05; Table 2). The large improvements against the best disorder predictor (AUC of 0.93 vs. 0.84 and AUPRC of 0.92 vs. 0.75) suggest that PSPHunter is well capable of accurately predicting phase-separating IDRs among both structured and disordered amino acids.

### 3.2. Prediction of Phase Separation in Disordered Residues

The second scenario predicts phase separation-associated IDRs among disordered residues on the low-similarity test dataset. Table 3, which summarizes these results, provides several practical insights. First, the disorder predictors are no longer capable of producing accurate results. Their AUROC and AUPRC values are not statistically different from the random baseline (*p*-value > 0.05). Nearly all phase separation predictors outperform both disorder predictors and are statistically better than the random baseline, which suggests that they can predict phase-separating IDRs. Several methods, such as Seq2Phase, PLAAC, ParSe v2, PScore, LLPhyScore, and catGranule, produce modestly accurate predictions, with AUROCs and AUPRC around 0.6 to 0.7. The most accurate results are again produced by PSPHunter, which obtained AUROC of 0.88 and AUPRC of 0.89; these results are significantly better than the results of the other considered tools (*p*-value < 0.05; Table 3).

Figure 1 provides further details by comparing side-by-side the tests on the disordered residues (second scenario) vs. disordered and structured amino acids (first scenario). We found that predictors provided either comparable or worse predictive quality when tested in the second scenario. Figure 1A sorts the methods by the magnitude of the differences between the two assessments and shows that flDPnn, AIUPred, FuzDrop, LLPhyScore, Seq2Phase, and PSPHunter tools produce statistically lower AUROCs in the second scenario (*p*-value < 0.05; blue bars in Figure 1A). The two disorder predictors that produce the largest magnitude of the differences, flDPnn and AIUPred, were designed to predict intrinsic disorder and not the phase-separating residues. As expected, they cannot discriminate between these different types of IDRs, resulting in poor quality in the second disorder-only scenario. Two more methods have large magnitudes of differences, FuzDrop (AUROC of 0.84 vs. 0.58) and LLPhyScore (AUROC of 0.82 vs. 0.62). FuzDrop was specifically designed to identify phase separation for the disordered proteins, and in ref. [15] it was similarly shown to underperform in the second scenario, securing AUROC of 0.50. Moreover, median Pearson correlation coefficients (PCC) between the FuzDrop ’s predictions and the outputs of the two disorder predictors for our test proteins that have IDRs that exclude the phase-separating regions (i.e., disordered proteins that lack phase-separating IDRs) are 0.74 (for AIUPred) and 0.65 (for flDPnn), suggesting that FuzDrop is biased to identify disordered regions as phase-separating. This bias can perhaps be explained by the fact that FuzDrop utilizes the disorder and disorder binding predictions as inputs [24]. LLPhyScore is also using disorder prediction as one of its inputs [23], and its median PCCs are 0.34 (with AIUPred) and 0.32 (with flDPnn) for the disordered proteins without phase-separating IDRs, suggesting a more modest bias to associate an overall intrinsic disorder with phase separation. The other two tools, Seq2Phase and PSPHunter, suffer smaller decreases in the AUROCs between the two scenarios, 0.81 to 0.69 for Seq2Phase and 0.93 to 0.88 for PSPHunter. Their corresponding median PCCs are small (0.21 and 0.15 for Seq2Phase and 0.02 and 0.10 for PSPHunter), which suggests that they are better equipped to differentiate between phase-separating IDRs and other types of IDRs. This is in line with their relatively high AUROC in the second scenario, at 0.69 for Seq2Phase and at 0.88 for PSPHunter (Table 3). Moreover, we found that several methods that managed to secure at least modest levels of performance did not record a noticeable difference in the performance between the two scenarios, including PLAAC, ParSe v2, and PScore.

Figure 1B compares the best predictor of phase separation, PSPHunter, against both baselines, the best disorder predictor in the context of identifying phase-separating IDRs, AIUPred, and the random predictor. This panel provides a holistic comparison across the four metrics of predictive quality using radar plots, i.e., lines closer to the outside denote higher predictive performance. We used dashed lines for the tests on the structured and disordered residues (the first scenario) and dotted lines for the tests on only the disordered residues (the second scenario). The red lines for the random baseline reveal that AIUPred (blue lines) cannot accurately identify phase separative IDRs among disordered residues, but it is much better than the random prediction across all metrics of performance when applied to the structured and disordered residues (the first scenario). This, again, is due to AIUPred’s ability to accurately differentiate between structured and disordered regions. More importantly, PSPHunter secured comparably high results on both tests (*p*-values of 0.48, 0.49, 0.08, and 0.03 for MCC, F1, AUPRC, and AUROC, respectively), which means that it provides accurate and robust predictions of the phase-separating IDRs.

### 3.3. Prediction of Phase Separation at the Protein Level

While we focus on the methods that target predictions at the amino acid level, they can also be used to identify proteins that are involved in phase separation. We computed the protein level predictions as the median of the amino acid level predictions and evaluated them on the 48 proteins from the low-similarity test dataset. We summarized these results in Table 4. We found that several predictors of phase separation can accurately identify the phase separation-associated proteins. In particular, the best results, with AUROCs > 0.84 and AUPRCs > 0.73, that are not significantly different (*p*-value > 0.05), are produced by FuzDrop, PSPHunter, and Seq2Phase. These high levels of predictive performance can be partly explained by the inclusion of the structured proteins in the test dataset, which should be easy to differentiate from the proteins with the phase-separating IDRs. This observation is supported by the high AUROC of 0.83 for the disorder predictor, AIUPred, which, as we demonstrated in Table 3, does not accurately distinguish between phase-separating and other types of IDRs. The other five phase separation predictors are statistically worse than FuzDrop and PSPHunter, which secure the highest AUROC and AUPRC values (*p*-values < 0.05), respectively. Altogether, while our analysis suggests that FuzDrop, PSPHunter, and Seq2Phase produce accurate protein-level predictions of the phase separation, this result should be considered with a certain amount of reserve, given the relatively small number of proteins in the test dataset.

## 4. Conclusions and Discussion

The protein sequence-based prediction of phase separation is an active research area [12,13,14,15], with over 30 methods and 9 that were published in 2024. We focused on the predictions of phase separation in IDRs and correspondingly, we covered all amino acid-level predictors. While a similar study was published in 2023, it was limited to just 5 methods and utilized predicted disorder as the ground truth [15]. We investigated eight amino acid-level predictors of phase separation using an experimentally annotated test dataset under two complementary scenarios and utilizing the low-similarity test dataset.

We found that several methods produce accurate predictions under the easier scenario that includes structured and disordered residues, with AUROCs ranging between 0.65 and 0.93. Moreover, modern disorder predictors also accurately predict phase-separating IDRs under this scenario, with AUCs as high as 0.84, matching or exceeding the performance of seven out of the eight phase separation predictors. Under the second and harder scenario that considers disordered regions, disorder predictors understandably underperform, and the majority of the phase separation predictors generate modestly accurate results (AUROC < 0.69). For comparison, the best of the five methods studied in ref. [15], LLPhyScore, obtained an AUROC of 0.67 in that study, and this method secured a similarly modest AUCROC of 0.62 in our tests.

While our results confirm the modest performance of the older tools that were studied in ref. [15], we identified a new and accurate predictor, PSPHunter [39], that secures AUROCs of 0.93 and 0.88 for the easier and harder scenarios, respectively. Table 1 reveals that PSPHunter relies on a modern machine learning model that uses a broad collection of predictive inputs, which include sequence embeddings, evolutionary conservation, and predictions of secondary structure, solvent accessibility, intrinsic disorder, DNA and RNA binding, selected posttranslational modification, and protein–protein interactions. These inputs express both the sequence patterns of the phase-separating IDRs (i.e., sequence embeddings) and their biophysical characteristics (i.e., high conservation, low propensity for secondary structure, and high propensity for binding and intrinsic disorder). Authors of PSPHunter stress the high predictive performance of certain inputs, such as the connectivity in the protein–protein interaction networks, evolutionary conservation, and abundance of certain posttranslational modifications [39]. Moreover, PSPHunter relies on the random forest model, which was shown to perform well across several related fields of research [77,78,79]. The use of the complementary inputs and the high-performing predictive model can explain why PSPHunter improves over the less accurate methods that typically apply simpler models and fewer inputs (Table 1).

We also highlight two predictors that produced relatively accurate results under the harder scenario, Seq2Phase and PLAAC, with AUROCs of about 0.68, neither of which were considered in the previous assessment [15]. Similar to PSPHunter, Seq2Phase [38] applies an ensemble of machine learning models, which include random forest, that utilize a wide spectrum of relevant inputs, such as hydrophobicity, the content of charged residues, the presence of low-complexity regions (which are a proxy for intrinsic disorder), predictions generated by Pscore [22], and sequence embeddings produced by the popular ProtTrans language model [80]. PLAAC utilizes a relatively simple Hidden Markov Model that relies on sequence-derived likelihoods derived from a collection of prion-like sequence regions [81]. Our results imply that the phase-separating IDRs share sequence characteristics with these regions. Moreover, our results suggest that two phase separation predictors, FuzDrop and LLPhyScore, tend to predict disordered regions as phase-separating, given their at least modest levels of correlations with the disorder predictions.

Lastly, we applied the eight amino acid-level methods to make predictions at the protein level. We identified three tools that produce accurate predictions of the phase-separating proteins: FuzDrop, PSPHunter, and Seq2Phase. However, this observation relies on a small collection of test proteins, and correspondingly, it should be taken with a pinch of salt.

We note that the fact that our study relies on sampling residues in the test proteins, which facilitates comparative evaluation across scenarios, means that it does not represent the diversity of results that can be produced across individual proteins. We showed that in the context of the assessment of the intrinsic disorder predictions [72]. We acknowledge that predictive quality will differ between proteins that have short vs. long phase-separating IDRs, that have large amounts of disorder, and that are fully structured. At this point, we could not provide a statistically sound analysis at the protein level, given the limited size of our test dataset. This will be a subject of our future work, once enough experimentally annotated test data becomes available.

## Figures and Tables

**Figure 1 biomolecules-15-01079-f001:**
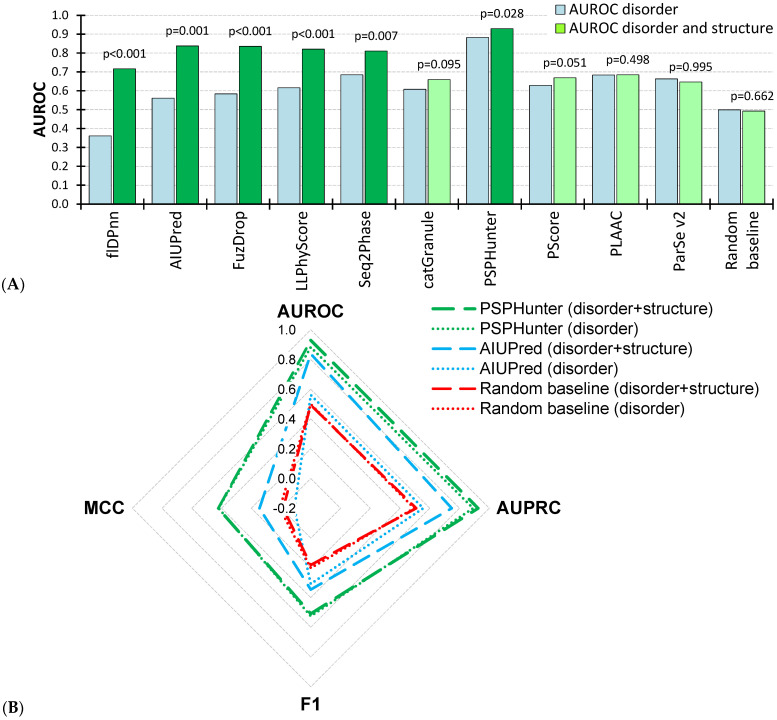
Comparison of results between tests that involved well-annotated disordered residues vs. well-annotated disordered and structured residues on the low-similarity test dataset. (**A**) reports mean values of AUROC over the 10 sampled subsets of the test dataset, respectively (blue bars for disorder-only test; green bars for disorder + structure test). We sorted methods by the magnitude of the differences between the two results and highlighted tools that obtained statistically significantly more accurate predictions on the disorder + structure dataset using a darker shade of green (*p*-value < 0.05; the corresponding *p*-values are at the top of the green bars). (**B**) gives the radar plot that compares AUROCs, AUPRCs, MCCs, and F1s between the best predictor of phase separation, PSPHunter (green lines), the best disorder predictor in the context of predicting phase separation, AIUPred (blue lines), and the random baseline (red lines). The Section 2.3 “Assessment of Predictive Performance” provides further details.

**Table 1 biomolecules-15-01079-t001:** Summary of the 31 phase separation predictors. Methods are sorted chronologically by their year of publication, and those in **bold font** satisfied the selection criteria, i.e., they make the amino acid-level predictions and had a working server or available code as of 22 June 2026 when the URL was accessed.

Method Name [Reference]	Publication Year	Has Server	Has Code	Code or Server Is Working	Provides Per-Amino Acid Scores	Brief Description of Predictive Model and URL(s) of Code and/or Server for the Selected Predictors
**PLAAC** [20]	**2014**	**Y**	**Y**	**Y**	**Y**	Hidden Markov Model that identifies prion-like domains and non-prion-like domains https://github.com/whitehead/plaac (code); and http://plaac.wi.mit.edu/ (server)
Baldwin et al. [40]	2015	N	N	N	N	
**catGranule** [21]	**2016**	**Y**	**N**	**Y**	**Y**	Scoring function that combines propensity for RNA binding, intrinsic disorder, and content of selected amino acids, which are processed using a sliding window of size 50 http://service.tartaglialab.com/new_submission/catGRANULE (server)
R + Y [41]	2018	N	N	N	N	
LARKS [42]	2018	N	N	N	N	
**Pscore** [22]	**2018**	**Y**	**Y**	**Y**	**Y**	Scoring function that considers short-range, long-range, backbone, and sidechain pi-contact predictions, which are processed using three sliding windows of sizes 40, 80, and 120 https://github.com/haocai1992/PScore-online (code); https://pound.med.utoronto.ca/JFKlab/Software/psp.htm (server)
PSPer [43]	2019	Y	Y	Y	N	
MaGS [44]	2020	Y	N	Y	N	
DeePhase [45]	2021	Y	Y	Y	N	
PSAP [46]	2021	N	Y	Y	N	
Droppler [47]	2021	N	Y	Y	N	
ParSe [18]	2021	Y	Y	Y	Y	There is a newer version of this tool that was published in 2023 and which we used for the assessment.
MaGSeq [48]	2022	Y	N	Y	N	
PASTA + Pscore [49]	2022	N	N	N	N	
**FuzDrop** [19,24]	**2022**	**Y**	**N**	**Y**	**Y**	Logistic regression that combines predicted probability for intrinsic disorder and for disordered binding https://fuzdrop.bio.unipd.it (server)
PhaSePred [50]	2022	Y	N	Y	N	
PSPredictor [51]	2022	N	Y	Y	N	
**LLPhyScore** [23]	**2022**	**N**	**Y**	**Y**	**Y**	Scoring function that uses eight predictive inputs: residue-water, residue-carbon, pi-pi, and short-range electrostatic interactions, hydrogen bonds, predicted short and long intrinsic disorder, and kinked beta-strands, which are processed with a sliding window of size 3 https://github.com/julie-forman-kay-lab/LLPhyScore (code)
dSCOPE [52]	2023	N	N	N	N	
PredLLPS_PSSM [53]	2023	Y	Y	Y	N	
PULPS [54]	2023	Y	N	Y	N	
**ParSe v2** [55]	**2023**	**Y**	**N**	**Y**	**Y**	Scoring regions in the 3-dimensional space defined by alpha-helix propensity, hydrophobicity, and v_model_ values, which are computed using a sliding window of size 25 https://stevewhitten.github.io/Parse_v2_web/ (server)
FLFB [56]	2024	Y	Y	Y	N	
Opt_PredLLPS [57]	2024	N	Y	Y	N	
PSPire [36]	2024	N	Y	Y	N	
Knowles et al. [58]	2024	N	Y	Y	N	
MolPhase [59]	2024	Y	N	Y	N	
PICNIC [60]	2024	N	Y	Y	N	
**PSPHunter** [39]	**2024**	**Y**	**Y**	**Y**	**Y**	Random forest model that uses amino acid composition, evolutionary conservation, predicted secondary structure, solvent accessibility, intrinsic disorder, DNA and RNA binding, and selected posttranslational modification, protein–protein interaction, and sequence embedding generated with word2vec method as inputs http://psphunter.stemcellding.org/ (server); https://github.com/jsun9003/PSPHunter (code)
CANYA [61]	2024	N	Y	Y	N	
**Seq2Phase** [38]	**2024**	**N**	**Y**	**Y**	**Y**	An ensemble of random forest, support vector machine, gradient boosted decision tree and shallow feedforward neural network that uses hydrophobicity, content of charged residues, and low-complexity regions, Pscore prediction, embeddings with ProtTrans model, and prediction of intrinsic disorder as inputs, with the results processed with a sliding window of size 100. https://github.com/IwasakiLab/Seq2Phase (code)

**Table 2 biomolecules-15-01079-t002:** Evaluation of the amino acid level predictions of phase separation-associated IDRs among well-annotated disordered and structured residues on the low-similarity test dataset. We reported mean values of the four metrics of predictive performance over the 10 sampled subsets of the test dataset. We highlighted the best score for each metric using bold font, and we sorted methods by their AUROC values. We showed results of tests of the statistical significance of differences in predictive performance in superscript next to the reported number using the *x*/*y* format where *x* compares a given method against the most accurate method and *y* against the random baseline, and where +, =, and − denote that a given result is statistically better, not statistically different, and statistically worse than the best/random predictor at *p*-value < 0.05. The Section 2.3 “Assessment of Predictive Performance” section provides further details.

		AUROC	AUPRC	F1	MCC
Predictors of phase separation	PSPHunter	**0.929** ^/+^	**0.925** ^/+^	**0.509** ^/+^	**0.423** ^/+^
	FuzDrop	0.836 ^−/+^	0.769 ^−/+^	0.311 ^−/+^	0.176 ^−/+^
	LLPhyScore	0.821 ^−/+^	0.808 ^−/+^	0.458 ^−/+^	0.306 ^−/+^
	Seq2Phase	0.811 ^−/+^	0.820 ^−/+^	0.278 ^−/+^	0.253 ^−/+^
	PLAAC	0.685 ^−/+^	0.775 ^−/+^	0.458 ^−/+^	0.395 ^−/+^
	PScore	0.669 ^−/+^	0.724 ^−/+^	0.456 ^−/+^	0.314 ^−/+^
	catGranule	0.660 ^−/+^	0.723 ^−/+^	0.445 ^−/+^	0.246 ^−/+^
	ParSe v2	0.647 ^−/+^	0.734 ^−/+^	0.442 ^−/+^	0.327 ^−/+^
Predictors of intrinsic disorder	AIUPred	0.838 ^−/+^	0.747 ^−/+^	0.346 ^−/+^	0.145 ^−/+^
	flDPnn	0.717 ^−/+^	0.646 ^−/+^	0.537 ^−/+^	0.273 ^−/+^
Random baseline		0.493 ^−/^	0.508 ^−/^	0.181 ^−/^	−0.014 ^−/^

**Table 3 biomolecules-15-01079-t003:** Evaluation of the amino acid level predictions of phase separation-associated IDRs among well-annotated disordered residues on the low-similarity test dataset. We reported mean values of the four metrics of predictive performance over the 10 sampled subsets of the test dataset. We highlighted the best score for each metric using bold font, and we sorted methods by their AUROC values. We showed results of tests of the statistical significance of differences in predictive performance in superscript next to the reported number using the *x*/*y* format where *x* compares a given method against the most accurate method and *y* against the random baseline, and where +, =, and − denote that a given result is statistically better, not statistically different, and statistically worse than the best/random predictor at *p*-value < 0.05. The Section 2.3 “Assessment of Predictive Performance” provides further details.

		AUROC	AUPRC	F1	MCC
Predictors of phase separation	PSPHunter	**0.882** ^/+^	**0.891** ^/+^	**0.525** ^/+^	**0.420** ^/+^
	Seq2Phase	0.685 ^−/+^	0.710 ^−/+^	0.267 ^−/+^	0.185 ^−/+^
	PLAAC	0.684 ^−/+^	0.747 ^−/+^	0.436 ^−/+^	0.363 ^−/+^
	ParSe v2	0.663 ^−/+^	0.722 ^−/+^	0.452 ^−/+^	0.324 ^−/+^
	PScore	0.629 ^−/+^	0.659 ^−/+^	0.455 ^−/+^	0.263 ^−/+^
	LLPhyScore	0.616 ^−/+^	0.660 ^−/+^	0.405 ^−/+^	0.168 ^−/+^
	catGranule	0.608 ^−/+^	0.694 ^−/+^	0.459 ^=/+^	0.137 ^−/+^
	FuzDrop	0.584 ^−/=^	0.627 ^−/+^	0.309 ^−/+^	0.008 ^−/=^
Predictors of intrinsic disorder	AIUPred	0.561 ^−/=^	0.556 ^−/=^	0.308 ^−/+^	−0.089 ^−/=^
	flDPnn	0.361 ^−/=^	0.435 ^−/=^	0.421 ^=/+^	−0.180 ^−/=^
Random baseline		0.499 ^−/^	0.495 ^−/^	0.201 ^−/^	0.009 ^−/^

**Table 4 biomolecules-15-01079-t004:** Evaluation of the protein-level predictions of phase separation-associated proteins among the disordered and structured proteins on the low-similarity test dataset. We reported mean values of the four metrics of predictive performance over the 10 sampled subsets of the test dataset. We highlighted the best score for each metric using bold font, and we sorted methods by their AUROC values. We showed results of tests of the statistical significance of differences in predictive performance in superscript next to the reported number using the *x*/*y* format where *x* compares a given method against the most accurate method and *y* against the random baseline, and where +, =, and − denote that a given result is statistically better, not statistically different, and statistically worse than the best/random predictor at *p*-value < 0.05. The Section 2.3 “Assessment of Predictive Performance” provides further details.

		AUROC	AUPRC	F1	MCC
Predictors of phase separation	FuzDrop	**0.882** ^/+^	0.737 ^=/+^	**0.812** ^/+^	**0.711** ^/+^
	PSPHunter	0.863 ^=/+^	**0.804** ^/+^	0.793 ^=/+^	0.694 ^=/+^
	Seq2Phase	0.848 ^=/+^	0.776 ^=/+^	0.756 ^=/+^	0.643 ^=/+^
	LLPhyScore	0.825 ^−/+^	0.676 ^−/+^	0.715 ^−/+^	0.567 ^−/+^
	PScore	0.738 ^−/+^	0.582 ^−/+^	0.491 ^−/+^	0.242 ^−/+^
	PLAAC	0.700 ^−/+^	0.722 ^−/+^	0.619 ^−/+^	0.424 ^−/+^
	catGranule	0.680 ^−/+^	0.516 ^−/+^	0.508 ^−/+^	0.255 ^−/+^
	ParSe v2	0.645 ^−/+^	0.666 ^−/+^	0.571 ^−/+^	0.346 ^−/+^
Predictors of intrinsic disorder	AIUPred	0.826 ^−/+^	0.602 ^−/+^	0.723 ^−/+^	0.574 ^−/+^
	flDPnn	0.680 ^−/+^	0.468 ^−/+^	0.582 ^−/+^	0.353 ^−/+^
Random baseline		0.484 ^−/^	0.351 ^−/^	0.220 ^−/^	−0.203 ^−/^

## Data Availability

The original contributions presented in this study are included in the article/Appendix A. Further inquiries can be directed to the corresponding author.

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
