# Peer review of "Empirical Assessment of Sequence-Based Predictions of Intrinsically Disordered Regions Involved in Phase Separation"

_biomolecules, 2025, doi:10.3390/biom15081079_

Round 1
Reviewer 1 Report
Comments and Suggestions for Authors
In this manuscript, the authors reviewed the performance of eight predictors of phase separation, which are capable of identifying phase-separating IDRs, on a well-annotated and low-similarity dataset. The results showed that the method PSPHunter is the most accurate tool for identifying phase-separating IDRs. Interestingly, specific methods designed for predicting IDR are not performing well on this dataset. Further, the authors provided a detailed explanation for the better performance of PSPHunter.
The manuscript is well written and data are presented in detail. It is acceptable for publication in Biomolecules
Author Response
Comment 1: In this manuscript, the authors reviewed the performance of eight predictors of phase separation, which are capable of identifying phase-separating IDRs, on a well-annotated and low-similarity dataset. The results showed that the method PSPHunter is the most accurate tool for identifying phase-separating IDRs. Interestingly, specific methods designed for predicting IDR are not performing well on this dataset. Further, the authors provided a detailed explanation for the better performance of PSPHunter.
REPLY: Thanks for your comment
Comment 2: The manuscript is well written and data are presented in detail. It is acceptable for publication in Biomolecules
REPLY: Thanks for your comment
Reviewer 2 Report
Comments and Suggestions for Authors
This manuscript presents a comparative analysis of residue-level phase-separation predictors for proteins. The methodology is sound and contains improvements relative to earlier related studies. The results are clearly described analyzed, and discussed. I have the following questions and remarks:
Some of the methods should be described more clearly:
- Training proteins: what were the training proteins, and how many? Did you obtain the list of training proteins used in the original publication of each method? If yes, describe this. There is, however, a caveat here, as the authors of a method may have retrained their method with more proteins since the publication of the method.
- Ensuring the low similarity between the training and testing proteins:
- Protein sequences were first clustered with BLASTCLUST. Describe how many sequences was the clustering applied to, and how many clusters you obtained.
- A "similarity threshold" of 0.3 was used for BLASTCLUST. I assume this is sequence identity. But it is not enough to only specify the sequence identity as it can be defined in many ways. What was the denominator? BLASTCLUST also allows the user to set criteria such as the sequences should match over what length of the sequences. Please specify the command line parameters you used with BLASTCLUST.
- Assessment of predictive performance. The text says: "We set the threshold for each predictor to the value that generates the correct number of phase-separating residues." Please explain this decision. In a real-life scenario, you don't know the correct number of phase-separating residues. Also, it is not clear whether the threshold was set separately for each test protein, or a single threshold applied to all test proteins.
Results:
- "we randomly sampled equal numbers of P and F residues in the second scenario, and the number of F and S residues was equal to the number of P residues for the first scenario" -- Please clarify that you sampled the outputs of the predictions, not the input sequences. Also, in the second scenario, clarify whether you meant F+S=P or F=P and S=P. Did the random sampling mean that the results from all test proteins were pooled?
- Inclusion of disorder predictors AIUPred and flDPnn: How did you use these to predict phase separation? Did you just consider all disordered residues phase separating? I don't see how this makes sense as your random sampling ensured that the number of F and P residues were equal, therefore a good disorder predictor will get about half of the predictions wrong (false positives). Also, although the disorder predictors seemed to perform well in the first scenario, it should be clarified that this would be worse in a real-life scenario where the fraction of P residues is probably small relative to all disordered residues (the performance will strongly depend on this fraction, which we don't know in advance for an individual protein).
- The disorder predictors were also included in scenario 2 where only the disordered residues were used. What is the point of this? In this case we expect these predictors to predict all residues as disordered (and "phase separating" in the test), so obviously they will fail.
- Section 3.1: "which suggests that they can accurately predict disorder from structure", I guess you meant "distinguish disorder from structure".
On another note, the fact that the authors randomly sampled the output of the predictors also prevents them from discussing the performance of the tested methods at the protein level, and on various types of proteins. Looking at the supplementary material, I see that there are several proteins where all residues are phase-separating while there are some where only a small region is phase-separating, and there are some where both P and F residues are present. Due to the random sampling, the proteins which only have short stretches of P residues will be underrepresented, and we don't know how well the predictors perform in predicting these regions. Presumably, the prediction is more difficult for these proteins than for those where the whole sequence is phase-separating. This limitation should at least be acknowledged.
Also, although the study is directed at predicting the phase separation property at the residue level, it would still be relevant to see how well these predictors perform at the protein level, e.g. if we consider a protein phase-separating if it is predicted to contain more than a predefined number of phase-separating residues.
I recommend this manuscript for publication after the above points are addressed.
Author Response
Comment 1: This manuscript presents a comparative analysis of residue-level phase-separation predictors for proteins. The methodology is sound and contains improvements relative to earlier related studies. The results are clearly described analyzed, and discussed.
REPLY: Thanks for your comment
Comment 2: Training proteins: what were the training proteins, and how many? Did you obtain the list of training proteins used in the original publication of each method? If yes, describe this. There is, however, a caveat here, as the authors of a method may have retrained their method with more proteins since the publication of the method.
REPLY: We obtained the training datasets of the phase separating proteins for the eight predictors from the supplementary materials (for catGranule, FuzDrop, LLPhyScore, ParSe, Pscore, Seq2Phase), website (for PSPHunter), and manuscript (for PLAAC). We collected 120 phase-separating proteins for catGranule, 453 for FuzDrop, 305 for LLPhyScore, 95 for ParSe, 28 for PLAAC, 11 for Pscore, 75 for Seq2Phase, and 167 for PSPHunter. We describe these details in Section 2.2. Our approach follows procedures used in equally well-executed studies in related areas of research. Note that, as we discussed in the introduction, past studies that evaluated phase separation predictors did not bother to remove similarity, and in one case, it was limited to 50%. We believe that the risk of authors retraining their models is negligible, as we are not aware of anyone doing that unless they publish a new version of their tool.
Comment 3: Ensuring the low similarity between the training and testing proteins:
Protein sequences were first clustered with BLASTCLUST. Describe how many sequences was the clustering applied to, and how many clusters you obtained.
REPLY: The revised text explains these details in Section 2.2. We clustered 1443 proteins and generated 703 clusters.
Comment 4: A "similarity threshold" of 0.3 was used for BLASTCLUST. I assume this is sequence identity. But it is not enough to only specify the sequence identity as it can be defined in many ways. What was the denominator? BLASTCLUST also allows the user to set criteria such as the sequences should match over what length of the sequences. Please specify the command line parameters you used with BLASTCLUST.
REPLY: We now provide details of how BLASTCLUST was used, including the command that we executed. We used the 0.3 similarity threshold and defaults for the other parameters. The key aspect of the defaults is that the high sequence coverage of 0.9 is used. We provide these details in Section 2.2.
Comment 5: Assessment of predictive performance. The text says: "We set the threshold for each predictor to the value that generates the correct number of phase-separating residues." Please explain this decision. In a real-life scenario, you don't know the correct number of phase-separating residues. Also, it is not clear whether the threshold was set separately for each test protein, or a single threshold applied to all test proteins.
REPLY: Good point. We extended the explanations of this aspect to provide the missing details. The revised text in Section 2.3 reads: “We set the threshold for each predictor to the value that generates the correct number of phase-separating residues across the entire test dataset (i.e., we used a single threshold for each predictor). We did that for comparative evaluation purposes to facilitate a consistent comparison between results produced by different methods, and to ensure that the binary state predictions are balanced. We note that individual predictors typically generate binary state predictions using their own thresholds that are optimized by the authors, and which should be utilized by the end users. However, such predictions could not be reliably compared across methods since they would result in vastly different rates of predictions of phase separating residues, which in turn would affect the ability to directly compare values of the considered metrics of predictive performance.”
Comment 6: "we randomly sampled equal numbers of P and F residues in the second scenario, and the number of F and S residues was equal to the number of P residues for the first scenario" -- Please clarify that you sampled the outputs of the predictions, not the input sequences. Also, in the second scenario, clarify whether you meant F+S=P or F=P and S=P. Did the random sampling mean that the results from all test proteins were pooled?
REPLY: Our description in this passage indeed missed several details that you pointed out. We have now expanded this explanation at the end of the first paragraph in Section 3 to address your comment. It reads as follows: “More specifically, we randomly sampled equal numbers of P and F residues in the second scenario, and the combined number of F and S residues was equal to the number of P residues for the first scenario, i.e., F+S=P. We used predictions from different methods for the same collection of sampled residues that were pooled across each of the 10 test protein sets, consisting of half of the test dataset, which we used to perform analysis of the statistical significance of differences.”
Comment 7: Inclusion of disorder predictors AIUPred and flDPnn: How did you use these to predict phase separation? Did you just consider all disordered residues phase separating? I don't see how this makes sense as your random sampling ensured that the number of F and P residues were equal, therefore a good disorder predictor will get about half of the predictions wrong (false positives). Also, although the disorder predictors seemed to perform well in the first scenario, it should be clarified that this would be worse in a real-life scenario where the fraction of P residues is probably small relative to all disordered residues (the performance will strongly depend on this fraction, which we don't know in advance for an individual protein).
REPLY: Thanks for this comment. We now detail the motivation for the inclusion of the two disorder predictors. The corresponding new passage at the end of the first paragraph in Section 3.1 says: “We utilize outputs generated by these two methods (i.e., their predictions of intrinsic disorder) as additional baselines that can be used to investigate whether the phase separation predictors would outperform the disorder predictors in the context of our study, which focuses on the intrinsic disorder. We note that the disorder predictors are expected to perform better than random based on scenario 1, where they should be capable of distinguishing phase separation-associated IDRs from structured residues. However, disorder predictors should perform only on par with a random baseline in scenario 2, given that they should not be able to distinguish phase separation-associated IDRs from other types of IDRs.” We also added an explanation for the results of these two methods in scenario 1 (middle of the second paragraph in Section 3.1) and in scenario 2 (middle of the second paragraph in Section 3.2). You are correct that the results in scenario 1 likely underestimate how well these disorder predictors will work in real-world (full sequence) prediction for disordered proteins, and we now discuss this in our revised text.
Comment 8: The disorder predictors were also included in scenario 2 where only the disordered residues were used. What is the point of this? In this case we expect these predictors to predict all residues as disordered (and "phase separating" in the test), so obviously they will fail.
REPLY: The point here is to show that disorder predictors are not somehow biased to find phase-separating IDRs and that, as expected, they fail in this scenario by being only on par with a random baseline. This highlights the value of the phase separation predictors. We explain the motivation to add these two predictors into our analysis in the new passage at the end of the first paragraph in Section 3.1
Comment 9: Section 3.1: "which suggests that they can accurately predict disorder from structure", I guess you meant "distinguish disorder from structure".
REPLY: You are correct. We fixed the wording per your suggestion.
Comment 10: On another note, the fact that the authors randomly sampled the output of the predictors also prevents them from discussing the performance of the tested methods at the protein level, and on various types of proteins. Looking at the supplementary material, I see that there are several proteins where all residues are phase-separating while there are some where only a small region is phase-separating, and there are some where both P and F residues are present. Due to the random sampling, the proteins which only have short stretches of P residues will be underrepresented, and we don't know how well the predictors perform in predicting these regions. Presumably, the prediction is more difficult for these proteins than for those where the whole sequence is phase-separating. This limitation should at least be acknowledged.
REPLY: Good points. The purpose of sampling was to facilitate direct comparisons across scenarios. We agree that our analysis does not represent the per-protein results, which would be different depending on whether these proteins include long vs short phase-separating IDRs, large vs small amounts of intrinsic disorder, or no disorder at all. We could not perform such an analysis given the limited size of our dataset. We now acknowledge and discuss this limitation in the last paragraph in the “Conclusions and discussion” section, and we point to this analysis as a potential future work.
Comment 11: Also, although the study is directed at predicting the phase separation property at the residue level, it would still be relevant to see how well these predictors perform at the protein level, e.g. if we consider a protein phase-separating if it is predicted to contain more than a predefined number of phase-separating residues.
REPLY: We considered and implemented your suggestion. We used the same test dataset and evaluation metrics/procedures to evaluate predictions at the protein level. However, we did not use the number of predicted phase-separating residues as the protein level score, and instead we opted to use the median value of the per-residue scores. The latter produces slightly better results. We describe and discuss these results in the new Section 3.3 and Table 4, and we briefly mention them in the second-to-last paragraph in the “Conclusions and discussion” section. We found three phase predictors that perform well, which include the best overall at the residue level PSPHunter method. However, we make a point that these results may not be reliable given the small number of proteins in our test dataset (in contrast to the number of amino acids that we used for the amino acid level assessments).
Comment 12: I recommend this manuscript for publication after the above points are addressed.
REPLY: Thank you. We believe we were able to address your comments.
Reviewer 3 Report
Comments and Suggestions for Authors
This manuscript benchmarks ten sequence-based predictors of phase separation-associated intrinsically disordered regions (IDRs) using a curated set of experimentally validated LLPS IDRs. The authors also assess these predictors on datasets of generic IDRs and globular proteins, providing a fair and valuable empirical evaluation. Overall, the study is technically sound, timely, and well-motivated. Only minor improvements are needed to enhance clarity, biological context, and presentation.
-
In the introduction, the authors could briefly highlight known features that distinguish LLPS IDRs from generic IDRs.
-
The threshold optimization approach is unclear. The authors mention setting the threshold to match the number of LLPS residues, but it is not evident how reproducible or practical this is for external users. More clarity or code guidance would help.
-
Please clarify whether any overlap exists between the test proteins and the training datasets used by the predictors.
-
The biological interpretation could be strengthened. For example, discussing why certain predictors (e.g., PSPredictor or metapredict) outperform others would provide valuable insight.
-
Figure quality should be improved. The font size in axis labels and legends is too small, and some panels (e.g., ROC/PR curves, LLPS score plots) would benefit from brief explanatory annotations.
Author Response
Comment 1: This manuscript benchmarks ten sequence-based predictors of phase separation-associated intrinsically disordered regions (IDRs) using a curated set of experimentally validated LLPS IDRs. The authors also assess these predictors on datasets of generic IDRs and globular proteins, providing a fair and valuable empirical evaluation. Overall, the study is technically sound, timely, and well-motivated. Only minor improvements are needed to enhance clarity, biological context, and presentation.
REPLY: Thank you for your evaluation.
Comment 2: In the introduction, the authors could briefly highlight known features that distinguish LLPS IDRs from generic IDRs.
REPLY: Good suggestion. We now briefly describe several factors/features that were discussed in the literature, together with the corresponding citations. You can find this in the second paragraph in the Introduction section. These features cover polymer scaling exponent, propensity for binding, enrichment in beta-turns, and depletion in alpha-helices.
Comment 3: The threshold optimization approach is unclear. The authors mention setting the threshold to match the number of LLPS residues, but it is not evident how reproducible or practical this is for external users. More clarity or code guidance would help.
REPLY: We extended the explanations of this aspect to provide justification and practical guidance. The corresponding revised text in Section 2.3 reads: “We set the threshold for each predictor to the value that generates the correct number of phase-separating residues across the entire test dataset (i.e., we used a single threshold for each predictor). We did that for comparative evaluation purposes to facilitate a consistent comparison between results produced by different methods, and to ensure that the binary state predictions are balanced. We note that individual predictors typically generate binary state predictions using their own thresholds that are optimized by the authors, and which should be utilized by the end users. However, such predictions could not be reliably compared across methods since they would result in vastly different rates of predictions of phase separating residues, which in turn would affect the ability to directly compare values of the considered metrics of predictive performance.”
Comment 4: Please clarify whether any overlap exists between the test proteins and the training datasets used by the predictors.
REPLY: We ensured that the test proteins share low levels of similarity with phase-separating training proteins that were used to develop the eight tested predictors. More specifically, we limit the sequence similarity to < 30%. We mention this in the last paragraph in the Introduction section, explain details of how it was done in Section 2.2, and briefly reiterate it when discussing results in Section 3.
Comment 5: The biological interpretation could be strengthened. For example, discussing why certain predictors (e.g., PSPredictor or metapredict) outperform others would provide valuable insight.
REPLY: While PSPredictor method was not included in our assessment, we now attempt to explain why certain methods, such as PSPHunter, PLAAC, and Seq2Phase perform well in this intrinsic disorder-centric assessment. The most accurate PSPHunter uses inputs that express both sequence patterns of the phase-separating IDRs (i.e., sequence embeddings) and their biophysical characteristics (i.e., high conservation, low propensity for secondary structure, and high propensity for binding and intrinsic disorder). This, combined with the use of a high-performing Random Forest model, can explain why PSPHunter improves over the less accurate methods that typically apply simpler models and less comprehensive inputs, as we show in Table 1. Seq2Phase, like PSPHunter, utilizes an advanced predictive model and a wide spectrum of predictive inputs that combine sequence patterns (i.e., modern protein language model) and several biophysical characteristics. Interestingly, PLAAC utilizes a relatively simple Hidden Markov Model that relies on sequence-derived likelihoods derived from a collection of prion-like sequence regions. Our results imply that the phase-separating IDRs apparently share sequence characteristics with these regions. We included this extended discussion in the third and fourth paragraphs in the “Conclusions and discussion” section.
Comment 6: Figure quality should be improved. The font size in axis labels and legends is too small, and some panels (e.g., ROC/PR curves, LLPS score plots) would benefit from brief explanatory annotations.
REPLY: We fixed the figure according to your suggestions.
Reviewer 4 Report
Comments and Suggestions for Authors
The manuscript "Empirical assessment of sequence-based predictions of intrinsically disordered regions involved in phase separation" by Wu et al. reports a systematic evaluation of computational predictors for identifying intrinsically disordered regions (IDRs) involved in protein liquid-liquid phase separation (LLPS). The authors constructed a dataset of 48 proteins with experimental annotations to compare 8 different methods, evaluating them under both an "easier" and a more "challenging" scenario. The results lead them to conclude that many existing predictors primarily identify general IDRs, with limited capability to specifically detect phase-separations. Among the compared methods, PSPHunter was shown to outperform others on this task. The authors attribute this advancement to its use of modern machine learning techniques that integrate a wide array of features.
As numerous methods have been developed and proposed, a consistent comparison and the resulting insights are crucial and beneficial for both developers and users. The manuscript is well-structured and provides high-quality information. I recommend it for publication after minor revisions.
Specific comments:
(1) Y-axis scale in Figure 1A:
The bar plot in Figure 1A, which compares AUROCs, has a Y-axis that starts at 0.3. Because bar charts convey information through the visual ratio of heights or areas, the current presentation could be potentially misleading to some readers. Since the statistical tests already clearly establish the significance of the differences, I suggest modifying the Y-axis to start from 0. This would reduce the potential for misinterpretation and stand by best practices for data visualization.
(2) table numbering:
There appear to be two separate tables labeled as "Table 1". Please revise the numbering to assign unique labels to each table and update the corresponding in-text mentions accordingly.
(3) dataset size:
The dataset prepared in this study seems relatively small. Is this size considered sufficient to draw general conclusions? If not, it would be helpful for the readers to include a brief discussion on the reasons for the limited dataset size (e.g. lack of high-quality annotations?) and the potential limitations this imposes on the study's conclusions.
(4) discussion on the top-performing method:
The conclusion states that the performance of PSPHunter is likely attributable to its use of various features. Is there any direct or indirect insight into which specific features contribute most significantly to this performance? If so, mentioning it could deepen the discussion and provide valuable guidance for future method development.
Author Response
Comment 1: The manuscript "Empirical assessment of sequence-based predictions of intrinsically disordered regions involved in phase separation" by Wu et al. reports a systematic evaluation of computational predictors for identifying intrinsically disordered regions (IDRs) involved in protein liquid-liquid phase separation (LLPS). The authors constructed a dataset of 48 proteins with experimental annotations to compare 8 different methods, evaluating them under both an "easier" and a more "challenging" scenario. The results lead them to conclude that many existing predictors primarily identify general IDRs, with limited capability to specifically detect phase-separations. Among the compared methods, PSPHunter was shown to outperform others on this task. The authors attribute this advancement to its use of modern machine learning techniques that integrate a wide array of features.
As numerous methods have been developed and proposed, a consistent comparison and the resulting insights are crucial and beneficial for both developers and users. The manuscript is well-structured and provides high-quality information. I recommend it for publication after minor revisions.
REPLY: Thank you for your evaluation.
Comment 2: Y-axis scale in Figure 1A:
The bar plot in Figure 1A, which compares AUROCs, has a Y-axis that starts at 0.3. Because bar charts convey information through the visual ratio of heights or areas, the current presentation could be potentially misleading to some readers. Since the statistical tests already clearly establish the significance of the differences, I suggest modifying the Y-axis to start from 0. This would reduce the potential for misinterpretation and stand by best practices for data visualization.
REPLY: We fixed the figure accordingly, i.e., the y-axis starts from 0.
Comment 3: table numbering:
There appear to be two separate tables labeled as "Table 1". Please revise the numbering to assign unique labels to each table and update the corresponding in-text mentions accordingly.
REPLY: Well spotted. We indeed numbered tables incorrectly. This revision fixes the numbering of the tables across the entire manuscript.
Comment 4: dataset size:
The dataset prepared in this study seems relatively small. Is this size considered sufficient to draw general conclusions? If not, it would be helpful for the readers to include a brief discussion on the reasons for the limited dataset size (e.g. lack of high-quality annotations?) and the potential limitations this imposes on the study's conclusions.
REPLY: Good point. We agree that our dataset is limited in size, and we discuss this aspect in the revised manuscript. In particular, we included the following passage in the last paragraph in Section 2.2: “Our test dataset has a similar size to the test data of 46 proteins from the ref. [15], the only published study that performed evaluations at the amino acid level, and fixes key issues of that test data, such as the use of predicted disorder as ground truth, and potentially high similarity to the training data [15]. While our assessment is at the amino acid level and the test set covers about 23 thousand amino acids, we acknowledge that the number of proteins included is rather limited, which can adversely affect the reliability of our observations. This limitation stems from the relatively small number of experimentally annotated phase-separating IDRs that are currently available in DisProt [37] and PhaSepDB [64], especially after we factored in the removal of test data based on similarity to the training datasets.”
We also include a caution concerning the protein level results that we included based on comments from reviewer 2 in Section 3.3, where we say: “this result should be considered with a certain amount of reserve, given the relatively small number of proteins in the test dataset.” We again stress this point in the “Conclusions and discussion” section, where we state, “However, this observation relies on a small collection of test proteins, and correspondingly, it should be taken with a pinch of salt.”
Comment 5: discussion on the top-performing method:
The conclusion states that the performance of PSPHunter is likely attributable to its use of various features. Is there any direct or indirect insight into which specific features contribute most significantly to this performance? If so, mentioning it could deepen the discussion and provide valuable guidance for future method development.
REPLY: Thanks for this helpful suggestion. We investigated the PSPHunter article and found that the authors performed an analysis of the importance of their predictive features in Suppl Figure 1. We correspondingly expanded the discussion of PSPHunter in the “Conclusions and discussion” section to enumerate and categorize its features and discuss their importance. We say: “Table 1 reveals that PSPHunter relies on a modern machine learning model that uses a broad collection of predictive inputs, which include sequence embeddings, evolutionary conservation, and predictions of secondary structure, solvent accessibility, intrinsic disorder, DNA and RNA binding, selected posttranslational modification, and protein-protein interactions. These inputs express both sequence patterns of the phase-separating IDRs (i.e., sequence embeddings) and their biophysical characteristics (i.e., high conservation, low propensity for secondary structure, and high propensity for binding and intrinsic disorder). Authors of PSPHunter stress the high predictive performance of certain inputs, such as the connectivity in the protein-protein interaction networks, evolutionary conservation, and abundance of certain posttranslational modifications [39].”